

# Integrating smartrock and seismic monitoring to investigate bedload transport dynamics during rapid increase of stages in ephemeral streams

Matanya Hamawi[1], Joel P. L. Johnson[2], Susan Bilek[3], Jens M. Turowski[4]. Jonathan B. Laronne[1,5]

[1]Department of Earth and Environmental Science, Ben Gurion University of the Negev, Beer-Sheva, 8410501, Israel
[2]Department of Earth and Planetary Sciences, University of Texas, Austin, 78712, USA
[3]New Mexico Institute of Mining and Technology, Socorro, 78712, USA
[4]GFZ Helmholtz Centre for Geosciences, Telegrafenberg, 14473, Germany
[5]Dead Sea and Arava Science Center, Masada National Park, 8691000, Israel

*Correspondence to*: Matanya Hamawi (hamawim@post.bgu.ac.il)

**Abstract.** Bedload transport dynamics during rapid increases of stage remain poorly constrained, particularly in ephemeral streams where such conditions are common. We combined two cutting-edge monitoring techniques – smartrocks and seismic measurements - to investigate bedload transport patterns during rapid increase of stage in two ephemeral channels with different morphologies. The later technique was used to characterize bedload activity through the Power Spectral Density (PSD) of recorded seismic signals. Our observations reveal three distinct stages of bedload response: (1) At shallow relative depth ($h/d_{84} \leq 0.9$), rapid increase of stage enhanced bed material activity compared to steady flow, with PSD ratios ($PSD_{rapid\ stage\ rise}/PSD_{steady\ flow}$) above unity and a higher prevalence of vibrational movement under rapid stage rise conditions relative to steady flow; (2) At intermediate relative depths ($0.9 \leq h/d_{84} \leq 2.5$), the rapid increase of stage effect on bedload activity diminished; (3) At greater relative depths ($h/d_{84} \geq 2.5$), bedload activity is once again enhanced during rapid increase of rise, with both seismic energy and particle motion exceeding values observed under steady flow conditions. The transitions between these stages occurred at similar relative depths in both channels despite their different morphologies, suggesting that channel roughness strongly influences how rapid stage rises affect bedload transport.

## 1 Introduction

Bedload transport exerts a crucial control on the dynamics of streams. It influences the patterns of erosion and deposition, the spatio-temporal variations in bed topography and texture as well as the design and implementation of in-stream engineering projects and river restoration. Although bedload transport has been extensively studied and formulated under conditions of quasi steady flow (e.g. Ancey, 2020; Luque & Beek, 1976; Meyer-Peter & Müller, 1948; Parker et al., 1982; Wilcock & Crowe., 2003), research addressing its dynamics under unsteady flow conditions remains limited. Such conditions occur across



a wide range of systems, such as during tidal surges in coastal and estuarine environments (Chanson et al., 2011; Khezri and Chanson, 2012), tsunamis (Kihara et al., 2015; Shafiei et al., 2016), glacial dam outburst (Clague, 2000) and reservoir breach (Aureli et al., 2023; Castro-Orgaz and Chanson, 2020; Sky and Chaudhry, 1989). It is particularly common in ephemeral streams within semi-arid and arid regions, where flash floods generate abrupt increases in water depth over short time intervals.

During flash floods, rapid increases in water depth can enhance turbulent velocity fluctuations and shear stress (Halfi et al., 2023; Thappeta et al., 2023), resulting in higher bedload transport rates compared to steady flow conditions (Halfi et al., 2023; Khezri and Chanson, 2012). Flume experiments demonstrated that bedload transport rates under turbulent, unsteady conditions can significantly exceed those observed under steady flow, with enhancement factors ranging from 1.4 to 4.4 (Lee et al., 2004; Sumer et al., 2003). Similarly, Meirovich et al. (1998), using measurements made in desert streams, demonstrated that

accounting for variable water surface slopes can lead to up to 1.6-fold increase in predicted bedload transport rates. Halfi et al. (2023) demonstrated that a rapid increase of stage can mobilizes bedload even in flow conditions where the shear stress calculated using conventional equations is below the critical threshold for motion. In particular, this can happen when a bore propagates over a dry bed in unarmored channels, where bedload flux may vary linearly with shear stress (Cohen et al., 2010). However, the depth-dependent nature of bedload transport during rapid increases of stage remained poorly constrained. Rapid

increases of stage are often unpredictable, infrequent, and short-lived, posing challenges for traditional bedload monitoring methods. This limitation highlights the need for innovative approaches capable of capturing the dynamics of bedload transport during rapid increases of stage.

In recent years, seismic methods have been developed to monitor fluvial environments. These advancements have the potential to greatly enhance our ability to measure bedload transport in field settings, including during rapidly changing hydrographs.

By capturing signals produced by particle interactions with the channel bed, through collisions (Tsai et al., 2012) or rolling (Luong et al., 2024), seismic sensors placed along riverbanks can provide continuous high temporal resolution measurements of transport activity. Building on these developments, previous studies have employed seismic analyses to investigate bedload transport processes in various environments, particularly in perennial rivers across mountainous regions such as the Alps (Antoniazza et al., 2023; Bakker et al., 2020; Burtin et al., 2011; Roth et al., 2014; Roth et al., 2016), the Himalayas (Cook et

al., 2018), and Taiwan (Chao et al., 2015; Hsu et al., 2011; Nativ et al., 2025). However, applications in ephemeral rivers have been limited (Dietze et al., 2019; Lagarde et al., 2021; Luong et al., 2024), and the method has not been specifically evaluated during rapid increases of stage.

Given the developmental stage of seismic methods for bedload transport monitoring, their application in natural river systems requires validation through independent and in-stream methods. Previous studies have employed a variety of approaches to

monitor bedload transport including bedload samplers (Bakker et al., 2020; Lagarde et al., 2021; Luong et al., 2024), pipe and plate impact sensors (Burtin et al., 2016; Lagarde et al., 2021; Roth et al., 2016) and hydrophones (Matthews et al., 2024). Yet, seismic observations have not yet been combined with 'smartrocks' – artificial or natural pebbles embedded with Inertial Measurement Unit (IMU). IMU are Micro-electro-mechanical Systems which generally contain three types of sensors: accelerometers, gyroscopes and magnetometers. Smartrocks allow direct investigation of bedload transport dynamics from the





Lagrangian perspective of the tracer grain (Alhusban & Valyrakis, 2021; Al-Obaidi & Valyrakis, 2021a; Pretzlav et al., 2020, 2021). While it is challenging to use smartrocks to monitor grain trajectories or applied forces (Al-Obaidi and Valyrakis, 2021a; Maniatis et al., 2020), their utility for identifying transitions between motion and rest throughout flow events has been successfully demonstrated in previous studies (Olinde and Johnson, 2015; Pretzlav et al., 2020, 2021).

Here, we study the onset of bedload transport under rapidly rising stage – bores – in ephemeral channels. We combine two
cutting-edge monitoring techniques, seismic monitoring and smartrocks, to investigate how rapid increase of stage affects bedload transport across different water depths in ephemeral streams. By simultaneously deploying both monitoring techniques in two ephemeral streams with different morphologies, we aim to develop a more comprehensive understanding of how rapid increase of stage influences bedload transport processes. This approach enables us to bridge crucial gaps in our knowledge of sediment transport under unsteady flow conditions and provides insights relevant to both theoretical frameworks and practical
applications in river management. In addition, we expand the number of field-based observations of rapid stage increase (Halfi et al., 2023).

## 2 Methods

### 2.1 Study sites

We monitored two gravel-bed ephemeral channels, Nahal (Wadi) Anim and Nahal Yatir (Fig. 1) during the winters of 2022 and 2023. Both sites are situated within the semi-arid climate of the northern Negev Desert, Israel, and drain the southern Hebron hills into the eastern Beer-Sheva Basin. These channels were chosen since they both feature rapid increase of stage during flow events, yet they possess differing grain sizes and morphological features. This allows us to examine whether bedload transport under rapid increase stage condition is influenced by the morphological characteristics of the river bed.
Nahal Anim has a catchment area of 35 km², an average longitudinal slope of 0.4%, and a channel width of 6 meters. The channel bed morphology of Nahal Anim is characterized by flats and bars, with a median grain size ($D_{50}$) of 14 mm, based on Wolman counts (see Supplement S1). The Yatir channel has a catchment area of 180 km², an average longitudinal slope of 1.3%, a channel width of 10 meters, and a median grain size ($D_{50}$) of 76 mm (Fig. 1). Based on the Nevatim station of the Israel Meteorological Service, located 5km eastward and 10 km southward of the Yatir and Anim sites, rainfall events occur
mainly from October to April, with a mean annual precipitation of 130 mm and 35 days with rain on average.

Pressure transducers (Levelogger 5, Solinst, ±5 mm) were installed in a stilling well located in the ephemeral channel beds, operating at a sampling rate of one sample every 2 minutes during the 2021-2022 season and one sample every 30 seconds during the 2022-2023 season. Barometric pressure compensation for the transducer data were performed using a reference transducer (Barologger 5, Solinst, ±5 mm) located at the Eshtemoa station (e.g. Cohen et al., 2010), situated 6 and 12 km from
Nahal Anim and Nahal Yatir, respectively. Using the compensated pressure data, we generated stage hydrographs for each event. Data were analyzed from start to end of each Anim flow event, but only to the shallow part of the comparatively long




Yatir recessions. These hydrographs were analyzed to identify rapid stage-rise periods. The rate of rise was calculated for each segment of the hydrograph where stage rose $\geq 0.5$ cm min$^{-1}$ during at least 2 min (Supplement S2), comparable to the minimal values (rate of 0.8 cm min$^{-1}$ for at least 1.3 min) reported in Halfi et al. (2023). Additionally, we compute the maximum rate of rise within each rapid increase of stage (Supplement S2).

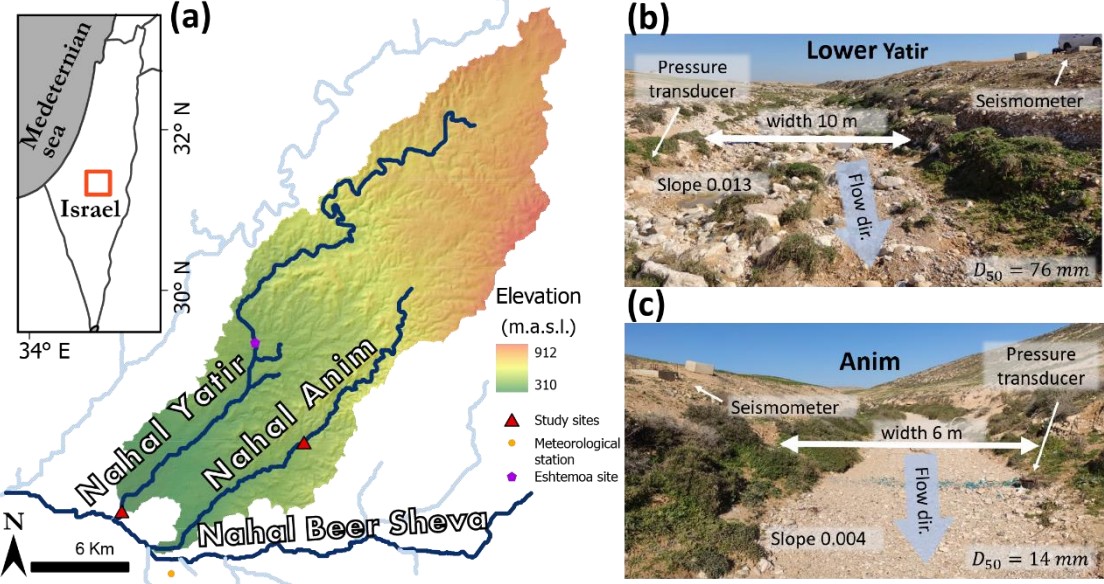

Figure 1. Study Sites: (a) Northern basin of Nahal Beer-Sheva with spatial location indicated in the inset. Upstream view of (b) Nahal Yatir site and (c) Anim site.

## 2.2 Instrumented Tracer Clasts

We used the same smartrocks as Pretzlav et al. (2020) to continuously monitor bedload movements. Each smartrock was equipped with an lMU (InvenSense 9150), set to record gyroscope (hereafter referred to as 'gyro') velocities at a sample rate of 10 Hz, able to measure up to ±2000° s$^{-1}$ at 16-bit resolution. The IMU was powered by a 3.7V, 5000mAh battery that enabled the smartrocks to function continuously for up to 30 days. After this deployment duration, data were downloaded and batteries replaced. To locate and identify the pebbles, an RFID tag was inserted into each pebble. All of the above components are housed in an artificial high-density plastic case, resulting into a bulk density of the smartrock of 2600 kg m$^{-3}$. The pebble has a triaxial ellipsoid-like shape, with axes length of 6.4 cm, 7.2 cm, and 13.0 cm. The intermediate diameter of 7.2 cm corresponds to grain percentile of $d_{95}$ and $d_{48}$ (where the subscript denotes the percentile of the distribution) in the Anim and Yatir sites, respectively. We placed the smartrocks on the bed, some exposed on the surface and others within imbricated sedimentological structures, along a 50-meter channel reach close to the seismometer (described in Sect. 2.2). Seven smartrocks were deployed at each site during the 2021–2022 season, and three were deployed during the 2022–2023 season.



The reduced number of smartrocks used in the latter season was due to the loss of some of them during flow events and damage caused by water infiltration.

We focused on gyro velocity data ($\omega_x, \omega_y, \omega_z$), because they are better indicators of movement than acceleration data (Al-
Obaidi and Valyrakis, 2021b). While Al-Obaidi & Valyrakis (2021b) found that roll, pitch, and yaw obtained through IMU data fusion are the best indicators of motion, such calculations require high sampling frequencies that reduce the operational duration of the smartrock and are generally challenging to implement (Maniatis, 2021).

All measurements recorded along the three axes were converted into the magnitude of gyro velocity using

$$|\omega| = \sqrt{\omega_x^2 + \omega_y^2 + \omega_z^2}. \tag{1}$$

For each measurement timestep, we selected the maximum magnitude value among all active smartrocks in the stream. The resulting time series was used to represent bed material movement during the event. To examine the relationship between smartrock gyro velocity and water depth, we calculated the median gyro velocity value within a 30-second window centered on each water depth measurement, i.e. 15 seconds before and 15 seconds after the water depth reading. The gyro velocity was then binned according to a 2 db logarithmic water depth interval.

Based on the gyro velocity data, we classified smartrock motion into three states: downstream displacement (hereafter referred to as *displacement*), vibration, and rest. The threshold value of 0.3 rad s$^{-1}$ distinguishing vibration from displacement was based on the laboratory measurements of Pretzlav et al. (2020)(Pretzlav et al., 2020), conducted with the same type of smartrocks. To determine the threshold between rest and vibration, we analyzed the data recorded during the period preceding the flow event. During this period, there was no water in the streams, and the smartrocks were expected to remain at rest unless disturbed by humans or animals. The starting point of these periods was defined as either the placement of the smartrocks in the stream
or the end of the previous flow event (when the channel was dry). For all data recorded during this time, we computed the envelope velocities based on maximum values for each minute. The threshold distinguishing rest from vibration was set to be very high - as the 99th percentile (≈2.3 standard deviations) of all computed envelope velocities during this period.

**2.3 Seismic Monitoring**

The study sites were equipped with 3-component broadband seismometers (Nanometrics Trillium Compact 120s sensors, with flat amplitude response between 120 s to >100 Hz) and a RefTek RT-130 datalogger sampling at 500 Hz and gain of 1, with an external GPS clock (GPS16X-HVS) attached for timing. The sensors were buried in the channel banks at a depth of 0.5 m, and at distances from the channel centerline of 10 m in Nahal Anim and 12 m in Nahal Yatir. The stations were powered using
12 V 100 Ah batteries. Stations were visited every three weeks to replace batteries and download data. Raw seismic data were archived with the EarthScope Data Management System under the network code 1F (Bilek, 2019).



### 2.3.1 Seismic data processing

We processed the raw seismic data (units of counts) first by removing the data mean and trend, then converted to velocity (units of m s$^{-1}$) using the station metadata (available from EarthScope for the 1F(2019-2024) network) and the
*remove_response* function contained within the ObsPy seismic analysis software (Beyreuther et al., 2010). Using the instrument corrected data, we calculated the Power Seismic Density (PSD) following the Welch (1967) method, with two-second time windows and a 50% overlap, in the frequency window of 10-100 Hz.

To investigate the seismic frequency range associated with bedload activity, we conducted a comparative analysis between gyro velocity and PSD. During each flow event, we calculated the median values of the PSD and gyro data in 30-second
intervals and then computed the Spearman's correlation coefficient between them. The gyro data used for this analysis were above the noise threshold, indicating motion (i.e., either downstream movement, or grain vibrations without downstream movement) and thus representing pebble-bed interactions that could generate seismic waves. This process was repeated multiple times, with each iteration selecting seismic energy within a 25 Hz frequency bin. For each successive iteration, the frequency window was shifted by 5 Hz relative to the previous one. For example, in one iteration, a frequency window of 10–
35 Hz was selected, followed by a window of 15–40 Hz in the subsequent iteration.

Frequencies with the highest Spearman correlation coefficients were selected to represent bedload activity. To assess changes in seismic response between rapid stage-rise and steady flow conditions, we analysed both the relationship between seismic energy and water depth, as well as between seismic energy and gyro velocity, using these bedload-indicative frequency bands. To reduce scatter and clarify trends, median values were computed for 2 dB bins based on water level and gyro velocity.


## 3 Results

### 3.1 Overview of Flow Events

During the measurement period, the sites experienced between two and five flow events per winter, with five events successfully recorded (Fig. 2), including one in Nahal Anim (hereafter termed *A1)* and four in Nahal Yatir (hereafter termed
*Y1-4*). Each event included multiple rapid increase of stage (25 rapid increase of stages recorded in total), with notable variability in their duration, magnitude, and rate of rise (Table 1; Supplement S3). In Nahal Anim, rise durations ranged from 5.5 to 27.5 minutes, with water depth increases of 4–23 cm. Average rates of rise varied between 0.5 and 0.9 cm min$^{-1}$ and maximum rates of rise ranged from 1.1 to 4.8 cm min$^{-1}$. In Nahal Yatir, durations of rapid increase of stage spanned 2 - 48 minutes, with corresponding water depth increases of 6 - 57 cm. The maximum rate of rise for an entire rapid increase was
17.8 cm min$^{-1}$; within individual rises maxima were 1.2 - 43.6 cm min$^{-1}$.





**Figure 2. Comprehensive data from the A1 (a) and the Y1–4 (b–e) flow events. The upper panels display water stage hydrographs, with yellow-shaded areas indicating intervals of rapid increase of stage. The lower panels present the spectrograms (left y-axis)**





alongside the gyro velocities of the smartrocks (right y-axis). The smartrock data are represented by a solid white line, with the rest-
to-vibration threshold indicated by a dashed white line.

**Table 1. Summary of hydrological data for the 25 rapid water level rises observed across the five flow events in Nahal Anim and Nahal Yatir. The rise numbers correspond to those depicted in Fig. 2. Rate of rise is the slope of a linear fit encompassing the entire rise range. Maximum rate of rise is the maximum slope within individual rises.**

| Event date | Event name | Rise number | Rise duration (min) | Water depth rise (cm) | Rate of rise (cm min⁻¹) | Max. rate of rise (cm min⁻¹) |
|---|---|---|---|---|---|---|
| 08/02/2023 | A1 | 1 | 5.5 | 0 - 4 | 0.8 | 1.6 |
| | | 2 | 8.5 | 5 - 15 | 1.3 | 2.1 |
| | | 3 | 14.5 | 1 - 8 | 0.5 | 1.1 |
| | | 4 | 8 | 9 - 14 | 0.8 | 1.6 |
| | | 5 | 11.5 | 8 - 19 | 1.0 | 2.0 |
| | | 6 | 27.5 | 20 - 43 | 0.9 | 4.8 |
| | | 7 | 16.5 | 3 - 18 | 0.9 | 1.9 |
| 27/01/2022 | Y1 | 1 | 4 | 0 - 6 | 1.5 | 1.6 |
| | | 2 | 8 | 9 - 66 | 8.5 | 27.3 |
| 04/02/2022 | Y2 | 1 | 18 | 0 - 9 | 0.5 | 1.2 |
| | | 2 | 4 | 9 - 44 | 8.8 | 10.3 |
| | | 3 | 6 | 46 - 74 | 4.9 | 6.7 |
| 26/12/2022 | Y3 | 1 | 6 | 0 - 9 | 1.0 | 10.1 |
| | | 2 | 2 | 5 - 35 | 17.8 | 43.6 |
| | | 3 | 12.5 | 38 - 51 | 1.1 | 2.2 |
| | | 4 | 3.5 | 36 - 82 | 13.0 | 31.5 |
| | | 5 | 21.5 | 85 - 94 | 0.5 | 1.9 |
| | | 6 | 16 | 84 - 115 | 1.4 | 14.9 |
| | | 7 | 7 | 56 - 75 | 2.9 | 4.3 |
| 08/02/2023 | Y4 | 1 | 13.5 | 0 - 55 | 2.6 | 61.4 |
| | | 2 | 16 | 32 - 41 | 0.7 | 1.3 |
| | | 3 | 21 | 35 - 65 | 1.9 | 5.6 |
| | | 4 | 20 | 40 - 67 | 1.4 | 4.4 |
| | | 5 | 14 | 64 - 88 | 1.9 | 3.3 |
| | | 6 | 7 | 46 - 75 | 4.8 | 8.9 |



### 3.2 Smartrock Dynamics

#### 3.2.1 Rest Periods

During rest periods without active flow, the gyro velocities of the smartrocks exhibited variability, with differences observed
in the velocity ranges recorded prior to each flow event (Fig. 3). The threshold velocity from motionless to in situ grain
vibrations (defined as the 99th percentile of measurements) ranged from 0.007 rad s$^{-1}$ prior to A1 (Fig. 3a-b) to 0.019 rad s$^{-1}$
prior to Y2 (Fig. 3e-f). Sporadic peaks exceeding 0.03 rad s$^{-1}$ were observed during the pre-flood periods, likely caused by
transient vibrations of the pebble resulting from animal bumps. The threshold value used to distinguish between rest and
vibrational motion was set to 0.007 rad s$^{-1}$ for Nahal Anim and to an event-average value of 0.014 rad sec$^{-1}$ for Nahal Yatir.

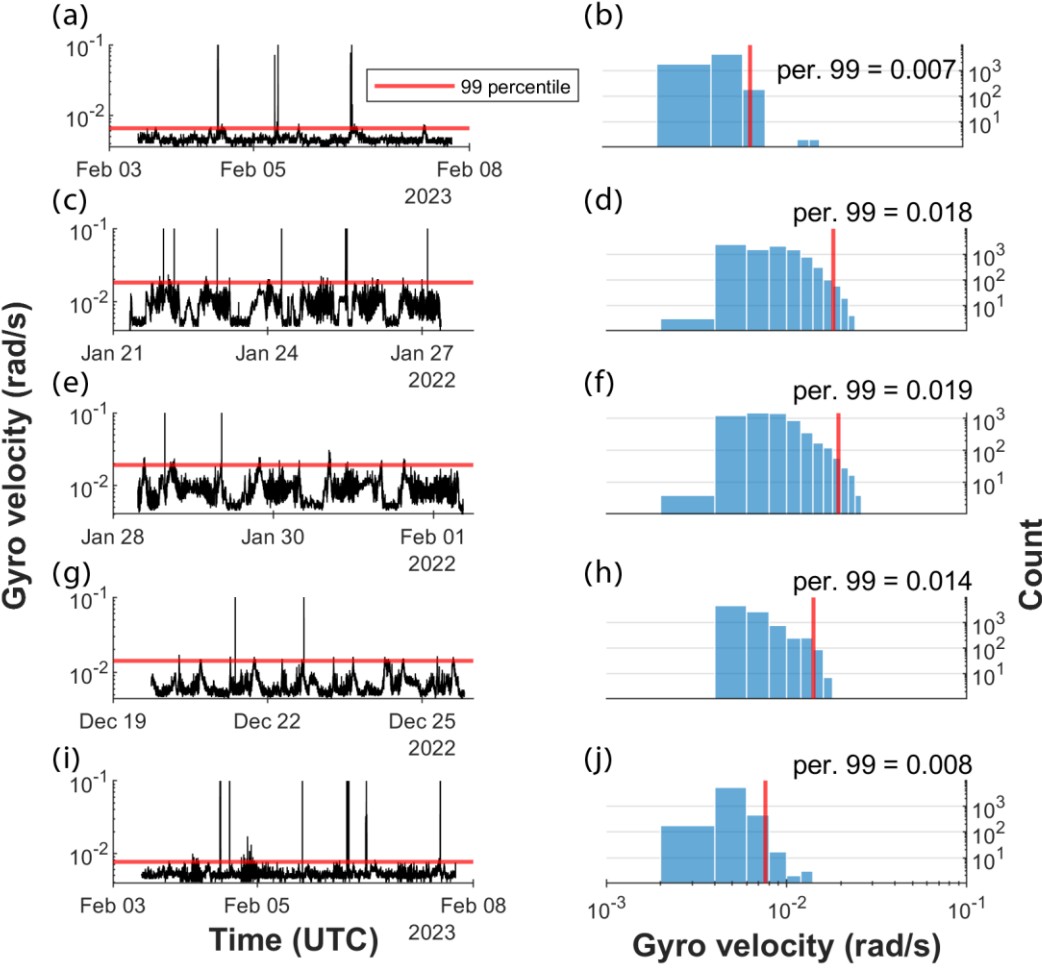


**Figure 3. Pre-flood gyro velocity time series (left side column) and the corresponding gyro velocity histogram (right side column).
The red line represents the 99th percentile value for each pre-flood period. (a-b) Nahal Anim; (c-d) Nahal Yatir, event 1; (e-f) Nahal
Yatir, event 2; (g-h) Nahal Yatir, event 3 (i-j) Nahal Yatir, event 4.**





### 3.2.2 Gyro Velocities during flow events

The smartrock gyro velocity time series recorded during flow events ranged from values below the noise threshold to peak rotation rates of 27 rad s$^{-1}$ and 16-39 rad s$^{-1}$ in A1 and Y1-4, respectively. The transition from velocities below the noise threshold to vibrational motion occurred at a water depth of 0.125 m in Nahal Anim and at depths of 0.25–0.40 m in event 1-4 in Nahal Yatir (Fig. S4), based on all gyro data. However, when separating measurements taken during rapid increase of stage from those taken under stable flow conditions, a distinct pattern emerges. During rapid increase of stage in Nahal Anim, gyro velocities exceeded the noise threshold at a water depth of ≈ 0.1 m while in steady flow condition it occurred at water depth of ≈ 0.2 m (Fig. 4a). In 6 out of 7 depth bins ≥ 0.1 m the median gyro velocities recorded during rapid increase of stage were higher than those observed under steady flow conditions. In Nahal Yatir, gyro velocities exceeded the noise threshold at depths in the range 0.10–0.30 m. When the gyro velocity exceeded the rest-vibration threshold in 13 among 14 depth bins, the median gyro velocities recorded during rapid increase of stage were higher than those observed under steady flow conditions. In Nahal Yatir the ratio of gyro velocity during rapid increase of stage to that during steady flow varies with increase in water level (Fig. 5). At depths up to ~ 0.09 m, this ratio exceeded unity; between ~ 0.09 and 0.65 m the ratio approached unity, except for an outlier at 0.24 m, after which the ratio again increased to values above unity. Based on these gyro velocity ratios, the data can be categorized into three distinct stages according to water depth (Fig. 5).







**Figure 4. The variation of smartrock gyro velocity with water level for rapid rise time windows (red) and steady time windows (blue). (a) Nahal Anim; (b-e) Nahal Yatir, events number 1-4. Dashed black line indicates the gyro threshold of average rest-vibration.**


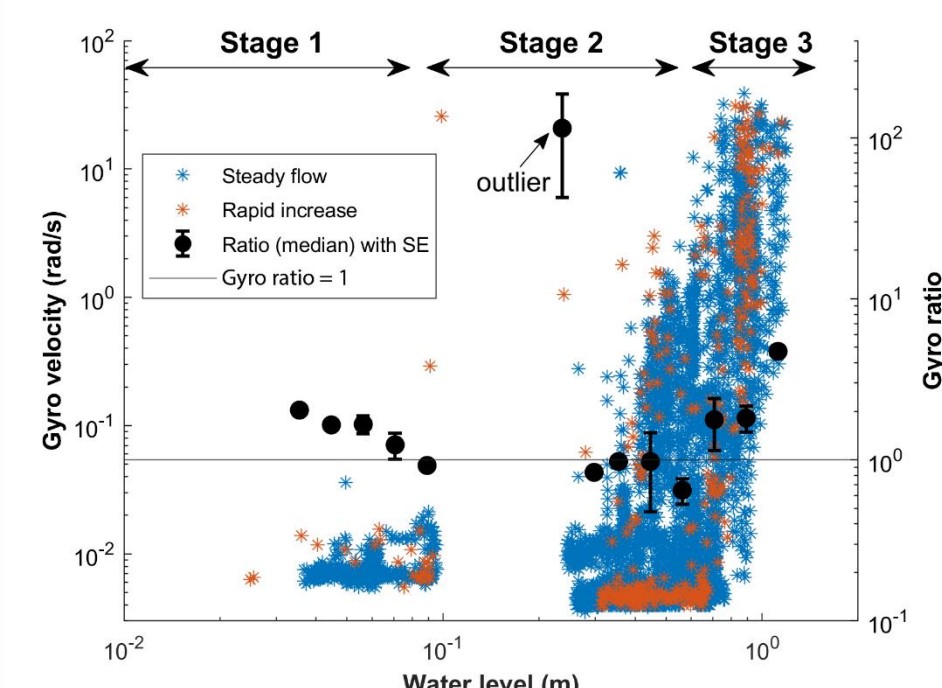

**Figure 5. Gyro velocity (left ordinate) and gyro velocity ratio (right ordinate) against water level from Nahal Yatir (events 1–4). The ratio points represent the median gyro velocity between rapid increase of stage and steady flow conditions, calculated for each water level interval. Based on the ratio points the data categorized into three distinct stages (black arrows). The horizontal line represents**
**a gyro velocity ratio of 1, indicating equal median velocities during rapid rise and steady flow conditions. Gyro data for the water depth range of 0.08–0.25 m is not shown, as this range corresponds to the very slow recession phase, where no comparable rapid stage rise gyro data exists.**

### 3.2.3 Distribution of smartrock movements

Figure 6 shows the distribution of smartrock movements relative to the total number of measurements, classified as either
vibration or displacement, across the three defined stages in Nahal Yatir. At this site the smart rock intermediate diameters are comparable to the median bed surface grain size ($\sim d_{50}$), making their movements representative of overall bedload transport. Smartrock motion during rapid increase of stage was considerably greater in stage 1 and 3, relative to steady flow. Additionally, when displacement occurred (stage 2 and stage 3), it was greater during rapid increase of stage compared to steady flow conditions.





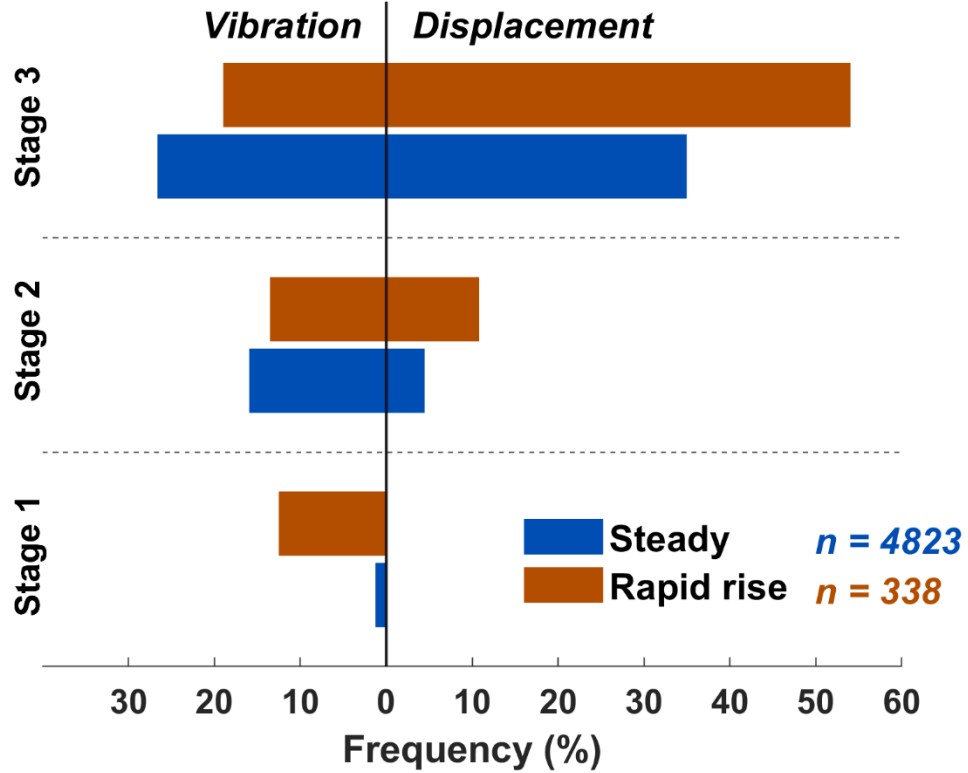


**Figure 6. Distribution of smartrock motion during rapid stage-rise and quasi-steady flow conditions in Nahal Yatir. Motion is categorized into vibration (left) and displacement (right) according to the three defined stages. Frequencies represent the proportion of measurements in each category relative to the total number of measurements, shown separately for rapid stage-rise (red) and steady flow (blue) conditions.**

**3.3 Characteristics seismic energy**

**3.3.1 Seismic Energy During Flow Events**

Seismic energy within the 10–100 Hz frequency range increased significantly during flow events, rising from noise levels of up to ~-180 db in Nahal Anim and Nahal Yatir to maximum values of -116 db and -120, respectively (Fig. 2). A distinct frequency range with elevated energy was observed during flow events, spanning 30-80 Hz in Nahal Anim and 40-90 Hz in

Nahal Yatir. The spectrograms did not reveal distinct or separate frequency bands that could be uniquely attributed to either turbulence or bedload transport. This observation suggests that these processes overlap and share common frequency ranges.

**3.3.2 PSD-smartrock gyro correlation**

Figure 7 presents the Spearman's correlation coefficients between median gyro velocity and median seismic energy across frequency bands for each event. In Nahal Anim, the frequency band with the highest correlation ($r = 0.82$) was 35–60 Hz

(average frequency 47.5 Hz). In Nahal Yatir, the frequency band with the highest correlation varied across events: 40–65 Hz,





60–85 Hz, 50–75 Hz, and 30–55 Hz for Events 2–5, with corresponding Spearman coefficients of 0.91, 0.86, 0.96, and 0.8, respectively.

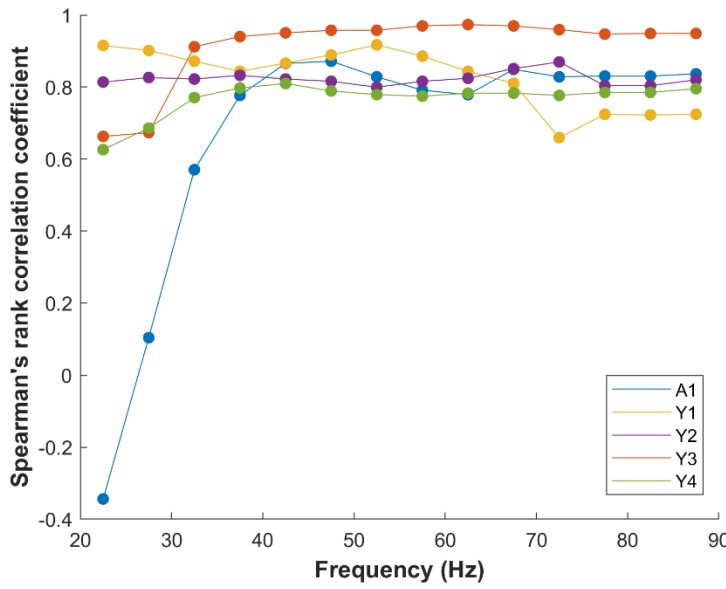

**Figure 7. The variation of Spearman's rank correlation coefficients for the PSD-gyro correlation with frequency. Each line**
**represents an individual event; circles indicate the mid-point frequency within each 25 Hz range.**

### 3.3.3 PSD trends with water depth

Seismic energy increased with water depth in Nahal Anim, exhibiting two distinct trends (Fig. 8). The first, a moderate increase, occurred up to a depth of approximately 0.10 m, with median PSD values (binned by water depth) ranged from -178
to -171 dB. The second, steeper increase extended to the maximum measured depth of 0.50 m, with median PSD values ranging from -170 to -145 dB in the second segment. In Nahal Yatir, PSD exhibited no significant trend with water depth up to 0.10 m, with bin median values ranging from -180 to -172 dB. At greater depths, seismic energy increased with water depth, ranging from -169 to -149 dB.







**Figure 8. PSD against water depth during A1, using 35-60 Hz (a); and Y 1-4, respectively using 40–65, 60–85, 50–75, and 30–55 Hz(b-e). Data points (grey crosses) are logarithmically binned (black circles) by water depth, with an interval of 2 dB relative to water level. PSD data for the water depth range of 0.08–0.25 m is not shown, as it corresponds to the very slow recession phase of this flow event, and no comparable rapid rise gyro data exists for this range.**

The ratio of seismic energy during rapid increase of stage to steady flow conditions varied with water depth (Fig. 9) at both sites. The PSD ratios exhibit a depth-dependent behaviour, enabling the data to be categorized into three distinct stages. While the general trends are consistent, the specific water depth ranges defining these stages differ between Nahal Yatir and Nahal Anim. In Nahal Yatir, Stage 1 occurs below ~ 0.1 m, where seismic energy ratios exceed unity. In Nahal Anim, this stage is observed at shallower depths, below ~ 0.03 m. Stage 2 spans ~ 0.1 to 0.55 m in Nahal Yatir and ~ 0.03 to 0.08 m in Nahal Anim, where seismic energy ratios approach or are slightly below unity. Stage 3 begins at depths greater than 0.55 m in Nahal Yatir and greater than 0.08 m in Nahal Anim, where seismic energy ratios increase once again. Despite these differences in absolute water depth, analysis of the combined dataset reveals a consistent pattern when examining the seismic energy ratio as a function of relative water depth ($h/d_{84}$; where $d_{84}$ represent one standard deviation above the mean roughness size) across both stations (Fig. 10). This unified analysis identified three distinct flow stages common to both channels. Stage 1 ($h/d_{84} \leq 0.9$) is characterized by ratios exceeding unity, Stage 2 ($0.9 \leq h/d_{84} \leq 2.5$) has ratios approximately equal to or slightly below unity, and for Stage 3 ($h/d_{84} \geq 2.5$) ratios consistently exceeded unity.

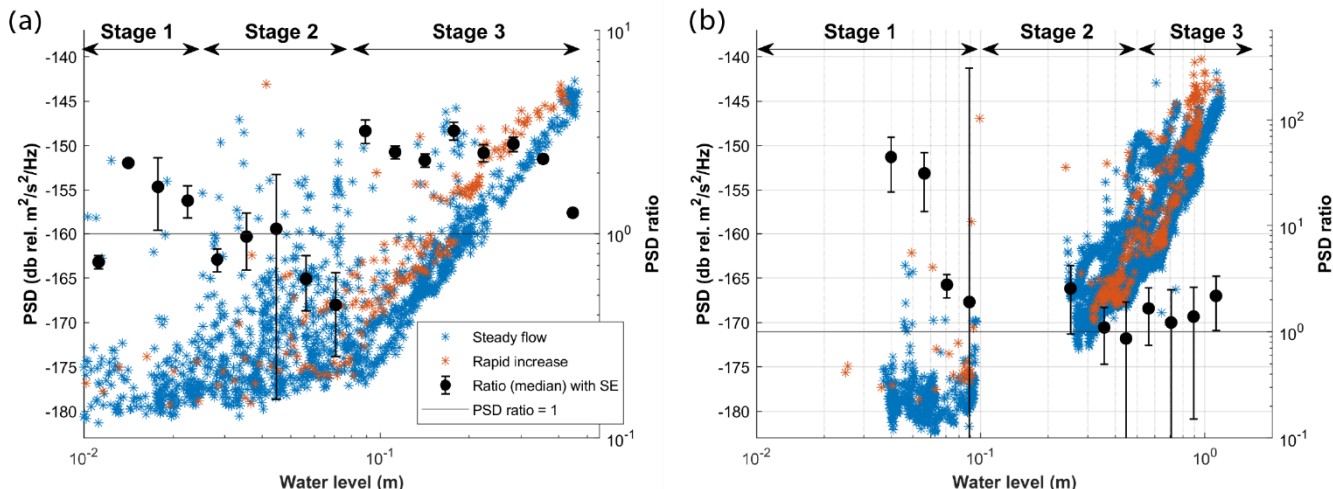

**Figure 9. The variation of PSD (left ordinate) and PSD ratio (right ordinate) with water level. The PSD ratio represent the median of PSD values between rapid increase of stage and steady flow conditions, calculated for each water level interval. (a) data from Nahal Anim and (b) from Nahal Yatir. The horizontal line represents a PSD ratio of 1, indicating equal median PSD during rapid rise and steady flow conditions. Based on the ratio points the data categorized into three distinct stages (black arrows). PSD data for the 0.08-0.25 m water depth range is not shown, as it corresponds to the very slow recession phase of this flow event, with no comparable rapid rise PSD data exists for this range. The seismic frequencies used here are those described in Sect. 3.3. The stages are very similar to those shown in Fig. 5, but not identical as two different methods are used.**




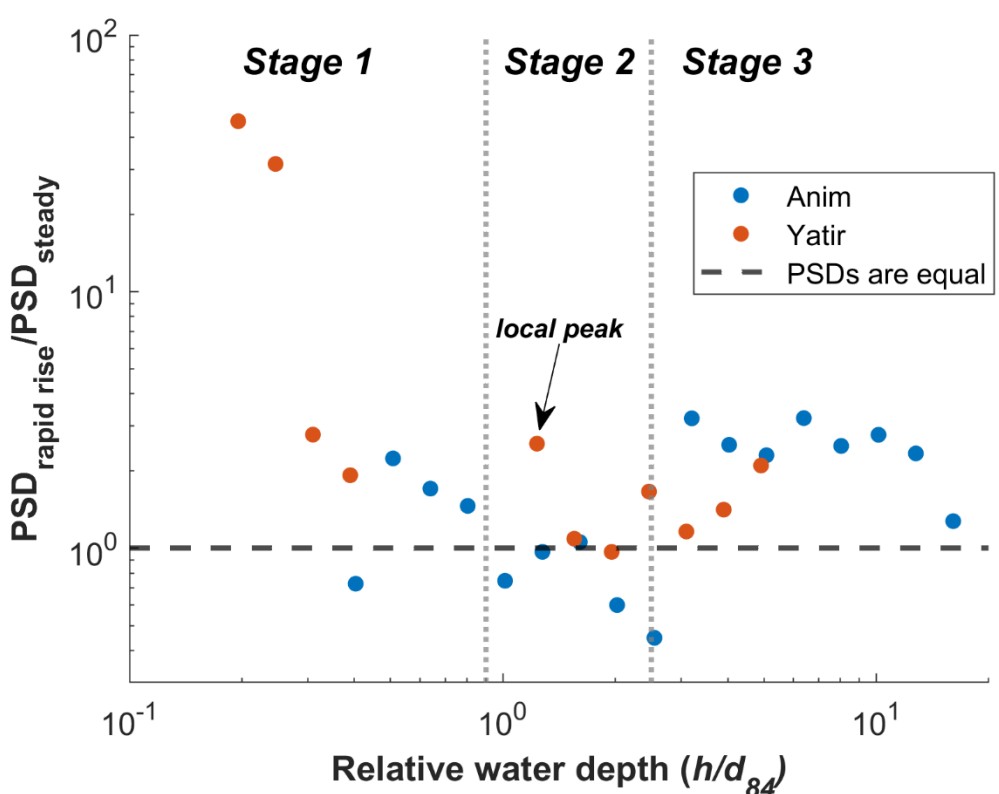

**Figure 10. PSD against relative water depth ($h/d_{84}$) for the single monitored Nahal Anim event (blue circles) and for Nahal Yatir, events Y1-Y4 (red circles). Dashed vertical lines indicate the transitions between stages.**

### 3.4 Coupling smartrock and seismic data

The relationship between the gyro velocity of smartrocks and seismic energies shows significant scatter below the noise threshold for all events, with seismic energy values in the range −183 to −142 dB for Nahal Anim (Fig. S9) and −189 to −148 dB for Nahal Yatir (Fig. 11 and Fig. S9), respectively. The median ratio of seismic energies under rapid stage-rise conditions to those under steady-flow conditions with increase of gyro velocity in Nahal Yatir remained almost consistently greater than

or equal to 1 across all velocities, but displayed distinct trends (Fig. 11). For velocities below 0.2 rad s$^{-1}$, the ratio *decreases* and varies between ≈1 and 3. At greater gyro velocities, the ratio *increases* in the same range.



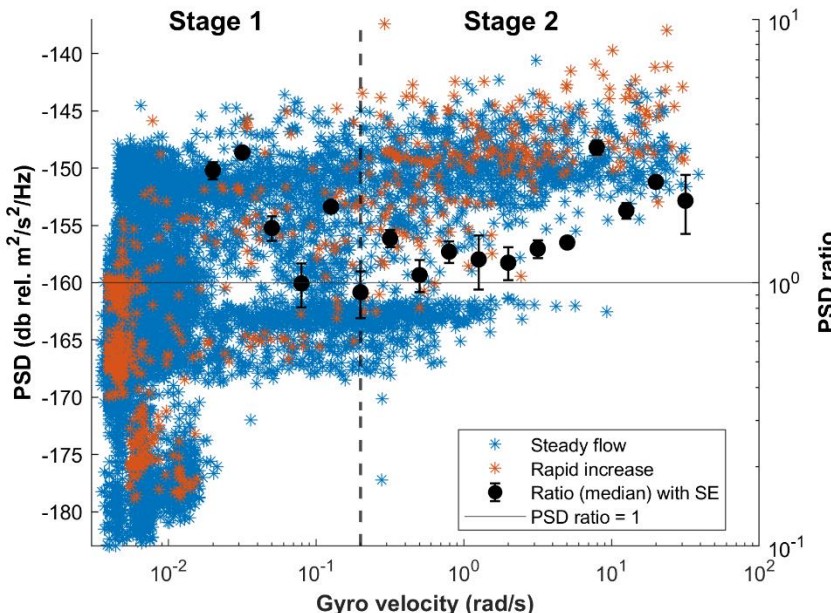

**Figure 11. The variation of PSD (left ordinate) and PSD ratio (right ordinate) with gyro velocity for Y1–4. The right ordinate represents the ratio of median PSD values between rapid increase of stage and steady flow conditions for each water level interval. Dashed line indicates transition between two stages. The horizontal line indicates a ratio of 1, denoting identical seismic energy for a given gyro velocity during rapid rise and steady flow conditions. The seismic frequencies used here are those described in Sect. 3.3.**

## 4 Discussion

To compare the bedload transport dynamics between rapid stage-rise and steady flow conditions, we collected seismic measurements and smartrock data in two ephemeral channels: Nahal Anim, characterized by finer bed material ($D_{50}$ = 14 mm), and Nahal Yatir, with coarser bed material ($D_{50}$ = 76 mm). We examined how gyro velocities varied with water depth, comparing rapid stage rise to steady flow conditions. Next, we analyzed seismic energy variations with water depth. Finally, we investigated the relationship between seismic energy and gyro velocities. The synthesis of these complementary measurement techniques revealed systematic patterns in how bed material responds to rapid increase of stage. In the following, we describe the stage characteristics in detailed, followed by a brief discussion of the potential use in seismic data inversion for calculating bedload transport rates during bores.



### 4.1 Stage Characteristics

Both the PSD ratio and the gyro velocity ratio independently revealed the existence of three distinct stages in the response of
bed material to rapid stage rises. These correspond to three stages in water level: 1) shallow, 2) transitional, and 3) deep. In
the following sections, we describe in detail and interpret the observations associated with each stage.

### 4.1.1 Stage 1: Shallow flow

At shallow relative depths ($h/d_{84} < 0.9$), seismic measurements reveal higher energies during rapid increase of stage
compared to steady conditions, extending to ~ 0.10 m in Nahal Yatir and ~0.03 m in Nahal Anim (Fig. 9). This observation is
also supported by the gyro velocity measurements in Nahal Yatir, where the gyro ratio between rapid increase of stage and
steady flow conditions remains above unity up to a depth of 0.09m (Fig. 5). These findings indicate that bedload transport
under rapid increase of stage at shallow depths is more pronounced than under steady conditions. This result aligns with
previous studies that have reported enhanced bedload transport due to higher turbulent velocity fluctuations (Halfi et al., 2023;
Sumer et al., 2003). These studies attribute this phenomenon to increased turbulence, associated with this flow condition, near
the streambed. Elevated near-bed turbulence increases instantaneous shear stresses on bedload particles, thereby amplifying
their transport.

Smartrock data for this stage indicate that under conditions of both steady flow and rapid increase of stage, the pebbles exhibit
no displacement, as gyro velocities remain predominantly below the motion threshold (Fig. 6). However, the pebbles do exhibit
vibrational motion, with substantially higher vibration occurrence during rapid increase of stage compared to steady flow
conditions (Fig. 6). This vibration behaviour is consistent with findings reported by Wang et al. (2023) that particle vibrations
are significantly influenced by high-energy turbulent events, which are crucial for particle entrainment. The elevated seismic
energy ratios observed at this stage in Nahal Yatir events, therefore, likely reflect bedload vibrations (Fig. 11), as evidenced
by gyro velocities being in the vibrational range ($0.015 < \omega < 0.3$ rad s$^{-1}$). However, it is plausible that a portion of the seismic
energy originates from the displacement of smaller pebbles, which constitute a minor fraction of the streambed material.

### 4.1.2 Stage 2: Transition


As relative depth increases ($0.9 \leq h/d_{84} < 2.5$), the seismic energy ratio diminishes, converging towards unity. This trend is
evident at water depths of ~0.10–0.55 m in coarser-grained Nahal Yatir and ~0.03–0.080 m in finer-grained Nahal Anim. In
the Nahal Yatir, where the smartrock approximates the $d_{50}$ of the bed material, gyro velocity measurements show
corresponding stage transitions at depths of ~0.09 m and 0.65 m. The reduction of the seismic energy ratio suggests that the
influence of rapid increase of stage on bedload transport decreases as water depth increases. A possible explanation is that
rapid stage increases over this range of water depths tended to occur with some water already flowing in the channel (Figure
2). Thappeta et al. (2023) found that hydraulic bores propagating over flowing water cause far lower bed shear stresses due to
turbulent velocity fluctuations compared to bores over dry beds. A similar trend was reported by Halfi et al. (2023), who found




that the impact of hydraulic bores on bedload transport rates is attenuated when it occurs over a wet bed compared to a dry
bed. They attributed this attenuation to the buffering effect of the pre-existing water layer, which mitigates the turbulence
generated by intense flow near the streambed. Their study further showed that this buffering effect strengthens with increasing
water depth, with the relative contribution of intense flow diminishing even before the threshold for bedload motion is attained.
During this stage the amount of vibrational motion under rapid increase of stage becomes nearly equal to that under steady
flow conditions, in contrast to the disparity observed in stage 1 (Figure 6). The smartrocks still exhibit predominantly
vibrational motion rather than displacement, with most gyro velocities generally remaining below the motion threshold.
Notably, the rest-to-vibration threshold values found in this study (0.007-0.015 rad s$^{-1}$) are identical to those we derived from
analyzing data presented in Al-Obaidi & Valyrakis (2021b), who investigated motion threshold characterization using
smartrocks (Supplement S6). Gyro velocities indicative of downstream motion are more frequent under rapid increase of stage
conditions compared to steady flow conditions, though these displacements remain relatively rare compared to vibrational
motion. In Nahal Yatir a notable peak in both gyro velocity and seismic energy ratios occurs at a depth of ca. 0.30 m (Fig. 5
and Fig. 9a), perhaps indicating the breakdown of some sedimentological structures before continuous movement begins.
Despite differences in absolute water depths between the two streams, the transitions between stages occur at similar relative
depths ($h/d_{84}$), suggesting that channel roughness may have a decisive influence on how rapid increases of stage affect bedload
transport. This observation aligns with the relative influence of turbulence associated with channel roughness, independent of
flow condition. As demonstrated by Lamb et al. (2008), turbulence fluctuations correlate with channel roughness and intensify
with increasing relative water depth. At stage 1, when water depth is shallow, during quasi-steady flow, turbulence intensity
is probably lower than when flows are deeper, causing turbulence from rapid increase of stage to be the dominant factor. As
depth increases, roughness-induced turbulence becomes more significant, reducing the relative contribution of rapid increase
of stage turbulence.

**4.1.3 Stage 3: Deep flow**

At greater relative depths ($2.5 \leq h/d_{84}$), both seismic energy and smartrock gyro velocity ratios increase yet again, suggesting
renewed influence of rapid increase of stage on bedload transport. The seismic energy ratio increases consistently with water
depth during this stage, reaching values well above unity (Fig. 9). Similarly, the gyro velocity ratio also shows an increasing
trend with water depth beyond 0.65 m in Nahal Yatir (Fig. 5). During this higher stage, motions and particularly displacements
are more frequent under rapid increase of stage conditions compared to steady flow (Fig. 6). This enhanced mobility is further
supported as the seismic energy ratio increases substantially at higher gyro velocities (above 1.2 rad s$^{-1}$), ranging from 1.4 to
3.3 (Fig. 11). Variations in seismic energies for identical gyro velocities could indicate that bedload transport occurs at varying
scales, with greater channel-wide transport observed during rapid increase of stage conditions compared to steady flow.



**4.2 PSD ratios as a proxy for bedload flux**

In this study, we utilized the ratio of seismic energies measured under two distinct flow conditions to investigate bedload transport dynamics in ephemeral streams. We propose that this parameter can serve as a proxy for the ratio of bedload fluxes under the following conditions: (1) seismic energy varies linearly with bedload flux within a specific frequency window (Luong et al., 2024; Tsai et al., 2012), and (2) the ratio is calculated using seismic energy within the frequency range associated with bedload transport. We used smartrocks to identify the bedload-related frequency range. However, since gyro velocity data

capture both vibrational motion and downstream displacement, the selected frequency range includes contributions from these two mechanisms. Consequently, the seismic energy used in the ratio calculation possibly includes a vibrational seismic source, thus, ratio values represent the upper bound of the actual bedload flux ratio.


**5 Conclusions**

Using both seismic and smartrock measurements, we observed three distinct stages in the manner by which intense flow conditions impact bedload transport dynamics. In shallow relative depths ($h/d_{84} < 0.9$), rapid increase of stage significantly enhances bed material activity (i.e. vibration or displacement) compared to steady flow conditions, with PSD ratios reaching

2-46 in Nahal Yatir and up to 2.2 in Nahal Anim, while gyro velocity ratios ($gyro\ velocity_{rapid\ stage\ rise}$/ $gyro\ velocity_{steady\ flow}$) in Nahal Yatir range from 1.2 to 2. At intermediate depths, this effect diminishes as channel roughness-induced turbulence becomes dominant, with both seismic energy and gyro velocity ratios approaching unity. As water depth further increases, measurements show enhanced transport during rapid increase of stage, with seismic energy ratios of 1.4-3.3 in Nahal Yatir and 2-3 in Nahal Anim, accompanied by elevated gyro velocity ratios in Nahal Yatir. The

correspondence between seismic and smartrock measurements validates these observations, providing a robust framework for understanding how flow conditions affect bedload transport.

The impact of rapid increase of stage on bedload transport highlights the importance of incorporating rapid increase of stage effects into bedload transport models. The elevated PSD ratios observed at low gyro velocities (Fig. 11) suggest that pebble vibrations significantly contribute to generation of seismic energy. This adds grain vibration as a source of bedload seismic

energy to grain impact (Tsai et al., 2012) and rolling (Luong et al., 2024). Vibrating grains may contribute substantial seismic energy, without contributing to bedload flux. As such, inversion methods aiming to back out bedload transport rates from riverine seismic signals (e.g., Dietze et al., 2019) may need to be adjusted for seismic noise due to vibrating grains, especially at low flow stages close to the threshold of grains motion.



**Data availability**

Seismic data collected is available through EarthScope under the 1F network code (2019-2024, Bilek, 2019). These services are funded through the National Science Foundation's Seismological Facility for the Advancement of Geoscience (SAGE) Award under Cooperative Agreement EAR-1724509.

**Author contribution**

Writing original draft - MH
Editing - MH, JPLJ, SB, JMT, JBL
Data analysis – MH, SB
Interpretation - MH, JPLJ, SB, JMT, JBL
Figures – MH

**Competing interests**

One of the authors is a member of the editorial board of Earth Surface Dynamics.

**Acknowledgements**

We are grateful to Yehoshua Ratzon for technical assistance and to Eitan Shemesh, Gal Wolfson, Tali Mushkovitz, Noam Mizrahi, and Idan Libman for fieldwork. Special thanks to Ron Nativ for his valuable suggestions. The seismic instruments
were provided by EarthScope Consortium through the EarthScope Primary Instrument Center at New Mexico Tech.

**Financial support**

This project was funded by the NSF-BSF program (grant 2018619). MH acknowledges support from the Kreitman School and the Department of Earth and Environmental Sciences at Ben-Gurion University of the Negev, as well as from the Fay and Rex Harbour Foundation.

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
