# Peer review of "Integrating smartrock and seismic monitoring to investigate bedload transport dynamics during rapid increase of stages in ephemeral streams"

_EGUsphere, 2025_

## Referee Comment (RC1)

**egusphere-2025-591**

**Integrating smartrock and seismic monitoring to investigate bedload transport dynamics during rapid increase of stages in ephemeral streams**

Matanya Hamawi, Joel P. L. Johnson, Susan Bilek, Jens M. Turowski, Jonathan B. Laronne

**General comments**

Dear authors, dear editor

This is a very interesting article on sediment dynamics in ephemeral streams. It is based on two different measurement techniques that are currently very popular in our field of research and which are combined here.

This parallel use of smartrock and seismic measurements is new and extremely exciting in bedload transport research. Quantifying seismic energy in the range where (artificial) bedload particles transition from vibration to actual displacement will certainly also be helpful for future studies using seismometers along torrents and mountain rivers outside semi-arid areas. The authors' focus on the moment in the discharge hydrograph when the water level/discharge rises sharply is clearly emphasised. Such rises occur, for example, during flash floods. Experts in the field are well aware that studying sediment transport dynamics during such rapid rises in discharge is of fundamental importance. This is because, compared to steady flow conditions, sediment transport rates are higher during the arrival of flash floods, when there is an intense and rapid increase in water levels. The fact that high sediment transport rates also greatly increase the risk of damage along watercourses could be explicitly mentioned (in the Introduction or Discussion sections) in order to also convince non-experts of the importance of this research.

I strongly support the publication of this article in Earth Surface Dynamics. It represents a step forward and will be important for the research community dealing with bedload transport. I can't think of any significant reasons why this manuscript shouldn't be published quickly. And below, I would like to list only two points of a more general nature.

However, as is well known, the devil is in the detail. This is evident in the considerable number of specific questions and comments that follow the more general remarks. However, I am confident that the authors will be able to respond to these comments and provide clarifications relatively easily, thereby strengthening their manuscript further.

[1] In the Introduction section (L44-46), the authors state that "Rapid increases of stage are often unpredictable, infrequent, and short-lived, posing challenges for traditional bedload monitoring methods." Further down the page (L59-61), several of these traditional measurements are listed in the text (samplers, impact sensors and hydrophones). In an innovative approach, the authors decided to use smartrock technology alongside seismic measurements. This combination is very well suited for improving the description of bedload transport dynamics during rapid increases in discharge in ephemeral streams.

However, in addition to a detailed description of the dynamics, it would also be interesting to be able to make quantitative statements about bedload transport rates or transported volumes during a bore. It might therefore be interesting for readers to learn briefly about the challenges or limitations of traditional methods of measuring sediment transport in semi-arid and arid regions especially in very unsteady flows. A reference to the relevant literature would be helpful. Perhaps to the work carried out in the nearby Nahal Eshtemoa (e.g. Halfi et al. 2018;

https://doi.org/10.1051/e3sconf/20184002036)? Could the seismic-smartrock combination also be used along Nahal Eshtemoa, where both direct and indirect sampling methods were employed? Provided, of course, that the slot samplers and pipe/plate microphones are still operational. I realise that there is not much space for this in the Introduction section of the article. I would therefore ask the authors to consider whether this point could be covered in a new, brief sub-section of the discussion.

[2] The authors introduce water level stages (1–3) for both monitoring techniques applied in this study. Stages 1-3 based on gyro velocity ratios are shown in Figure 5 (and introduced in the text in lines 213-214). Stages 1-3 based on PSD ratios are shown in Figure 9 (and introduced in the text in line 272). The only place in the Results section where the authors point out that there are two differently defined sets of stages (1–3) is in the caption of Figure 9, lines 289–290.

To avoid confusion among readers, I strongly recommend that these two differently defined sets of water level stages (1 to 3) also be named/labelled differently. This is all the more important given that yet another set of stages is introduced in Figure 10 (and in the text on line 280). These are also defined based on the PSD ratio. However, they refer to the relative water depth and apply to both study sites.

Figure 11 concludes the presentation of the results by showing how the PSD varies with smartrock gyro velocity. Here, the authors once again introduce stages (this time two, Stage 1 and Stage 2). These have no direct connection to the set of "stages" mentioned above. They should definitely be designated differently.

I do realise that the authors clarify this right at the start of the Discussion in section 4.1 and distinguish between the three stages determined using the PSD ratio and gyro velocity ratio. Nevertheless, I would like to insist on my comment regarding the need for greater clarity in the Results section. I think the manuscript would benefit from this.

In summary, I would be pleased if this article could be published in ESURF. I suggest the manuscript be re-evaluated following moderate revisions.

**Specific comments**

There is a somewhat confusing mix of upper and lower case letters in the section and subsection titles. I am not sure if there is a system behind this and recommend reviewing it again and applying a consistent approach.

- L48: In my opinion, this statement in the first sentence of the paragraph should be reinforced with a literary citation. Is there a review article on using seismic methods to monitor fluvial processes that could be cited here as an introduction?

  Burtin et al. (2016) comes to mind. It's been a while, given the speed of progress, but perhaps it would be appropriate here. [https://doi.org/10.5194/esurf-4-285-2016]
- L50-52: Regarding the statement made in this sentence (and also in the next one) about the continuous measurement of transport activity using seismic sensors: What is the limit of the grain size that can typically be detected? Are there any indications from studies in which these methods are used? This could be briefly added here.
- L52-55: Consider including examples from North America, such as the work of Schmandt et al. (2017) on the Trinity River in northern California. [https://doi.org/10.1130/G38639.1]
- L71-73: This sentence contains many repetitive elements that have already been mentioned in the sentence immediately preceding it. It should be revised, and perhaps the entire final paragraph could be made a little bit more concise.
- L74-75: Regarding insights relevant to practical applications in river management: can the authors give any concrete examples?
- L75-76: I'm not sure if I understand why the authors make this final reference to the literature in the introduction. Are Halfi et al. (2023) suggesting that more field-based observations of rapid stage increases in ephemeral streams should be carried out? I am not sure if all ESURF readers are aware of this. Perhaps it should be formulated a bit more clearly.
- L80-81: Perhaps clarify whether the authors mean the winter of 2021–22 or 2022–23 by "winter of 2022" (ditto for "winter of 2023" one line below). This is explained below (L92–93), but it would be better if this were made clear at the beginning of this subsection.
- Regarding the channel width: is this a typical, average, maximum or minimum value?

  Please specify, as the width is probably not identical along the entire length of the river channel. The same applies to the width specification for Nahal Yatir in line 88.
- L88-90: "Nevatim station of the Israel Meteorological Service": Is that the orange dot in Figure 1a? For clarity's sake, I suggest including the full name in Figure 1 (or in its caption).
   The kilometer specifications ("5 km eastward", "10 km southward") are not clear to me. Please check the wording.

Is the precipitation from October to April mainly advective, or is there also convective precipitation? This question is not entirely unimportant, as there is a considerable distance between the precipitation measurement and the areas of runoff formation of the two rivers.

- L95-96: Were there minimum criteria for considering a discharge event? Was every event during which water flowed in the channel used? Or, to put it another way: how was an event defined?
- L105: The term "tracer clasts" in this subtitle appears only here and nowhere else in the text.

  This should be reconsidered. Is there a reason why "smartrocks" or perhaps "pebbles"\* was not used?
- L125: Out of curiosity: how were the different smartrocks synchronised in terms of timing? I leave it up to the authors to decide whether this might be worth including as an informative sentence in this paragraph.
- L131-138: From the text, I gather that the threshold value used to distinguish between vibration and displacement (0.3 rad/sec) is the same for all smartrocks and for all flow events in both study streams. However, it is not clear to me whether the same applies to the threshold between rest and vibration.

  It would therefore probably be helpful to specify in subsection 2.2 if this lower threshold is (i) the same in both study sites, (ii) the same for the initial seven smartrocks in each study site and (iii) the same for the five flow events?
- L147: The numbering 2.3.1 does not make sense to me (as there is no subsection 2.3.2). Perhaps you could change it to "2.4 Seismic data processing" or just delete it and change subtitle 2.3 to "Seismic monitoring and data processing".
- L156-157: At lines 130–131, the authors define three states of smartrock motion, including "downstream displacement" (or simply "displacement") and "vibration" However, other terms are used here, such as "downstream movement" and "grain vibrations". Is there a reason for this, or could it be brought into line?
- L170: The authors should carefully consider whether they would like to introduce an abbreviation for the term "rapid increase of stage", which is omnipresent in the text. It might not be a bad idea, as this term appears in several sentences throughout the text alongside the later introduced "Stages 1-3" (terminology for water level ranges).

  An alternative would be to use "rapid water level rises" as in the caption of Table 1 (interestingly, this term only appears in this caption). Or both terms could be used as synonyms (which might then need to be defined).
- L171-175: Please check the values in the text carefully and compare them with those in Table 1. I have identified two discrepancies, or at least that is how it appears to me.

  L173 → According to Table 1 the longest duration of rise is 21.5 min (and not 48 min)

  L175 → And Y4 (rise no. 1) has a max. rate of 61.4 cm/min (not 43.6 cm/min)
- L190-191: Please use the three states of smartrock motion defined in lines 130–131, or explain why you are using different terminology.
- L193-194: My previous question regarding (L131-138) is answered here. However, I wonder: does it make sense to work with different thresholds for the two study sites? The sensors are the

same, after all. Perhaps the authors could briefly explain their rationale (here or in the *Methods* section).

L201-202: Consider extending to

"ranged from values below the rest-to-vibration (noise) threshold to peak rotation rates" Also, the peak rotation values mentioned in this opening sentence are shown in Figures 4 and 5. Shouldn't they also be visible in Figure 2 (solid white line in the lower panels)? Somehow, however, the peak values seem to be lower there.

- L212: The authors state in the caption of Figure 5 that gyro data at a water level between 0.08 and 0.25 m is not shown. Either change the value of 0.09 m at the beginning of the sentence on line 212 to 0.08 m, or edit the caption accordingly ("0.09-0.25 m"; the same would apply to the values given in the captions of Figures 8 and 9).
- L232: Regarding the use of "greater" in this line and the line below: do the authors mean "more frequent"?
- L258-263: In my opinion, it would be helpful to note here that the individual PSD data points for Nahal Anim (the grey crosses in Fig. 8a) are much more scattered than the corresponding data points for Nahal Yatir (Fig. 8b-e). This is particularly the case for Nahal Anim data in the water level range up to about 0.1 m. Is there an explanation for this?
- L331-332: One could think that the statement made in this sentence contradicts with the one made by the authors in line 337-338. Namely, that (according to smartrock data) "under conditions of both steady flow and rapid increase of stage, the pebbles exhibit no displacement" (in stage 1).

  However, it may well be that particles smaller than smartrocks already move in stage 1, meaning that bedload transport probably occurs under these conditions. Would it be worth mentioning this in this subchapter?
- L333-335: It seems to me that there are repetitive elements in these two sentences. This could be optimised.
- L346-349: In the first four lines of subsection 4.1.2, I miss references to figures in the text. I am not entirely sure whether this is an oversight or intentional on the part of the authors (if the latter is the case, please explain). I suggest the following additions:

L346: "... converging towards unity (Fig. 10)."

L347: "... in finer-grained Nahal Anim (Fig. 9)."

L349: "... at depths of ~0.09 m and 0.65 m (Fig. 5)."

- L351-352: Regarding the reference to Figure 2: Can any specific part of the graphic be referred to? Except for Figure 2(e) and perhaps 2(d), I find it difficult to follow the point the authors are making here.
- L354: What exactly do the authors mean by a 'wet bed'? A submerged channel bed or a channel bed that is saturated with water (but without a layer of water on top)?
- L364: The statement "though these displacements remain relatively rare compared to vibrational motion." could be supplemented with a reference to Figure 6.

- However, looking at Figure 6, I would say that this mainly applies to steady flow conditions and less so to rapid rise conditions.
- L365-366: The reference to Figure 9a is incorrect; Figure 9b is correct. Also, according to the two graphs, the indication of the water level is more likely to be 0.25 m (instead of 0.3 m).
- L367-369: Consider supporting the statement in the first part of this sentence with a reference to Figure 10.
- L377-378: I do not see this "consistent increase" in the PSD ratio during stage 3 in either Fig. 9a or Fig. 9b. Please consider rewording.
- L379-380: Figure 6 shows that the statement made in this sentence only applies to the "displacement" state. For the "vibrations" state, the opposite seems to be true (vibrations occur more frequently in stage 3 under steady flow conditions than during rapid rises).
- L384: The title of this subchapter is not entirely accurate. The authors suggest using the PSD ratio as a proxy for the ratio of bedload fluxes under two discharge conditions (rapid rise vs. steady flow). Is that correct?

  Perhaps the authors could briefly mention in this subsection what direct benefits this bedload flux ratio could have for research and/or practice.
- L397: I wonder if the formulation "... in the manner by which intense flow conditions impact bedload transport dynamics" is really suitable. After all, the shallow flow conditions in stage 1 (subsection 4.1.1) do not appear to generate any substantial discharge yet. As the name suggests, low water levels usually indicate low discharge.
- L399: Regarding the parenthesis in the middle of the line: "(i.e. vibration or displacement)"

  Doesn't Figure 6 show that there is no displacement in stage 1 (neither for rapid rise nor for steady flow conditions).
- L404: I cannot reconstruct these numbers.

  Nahal Yatir: I can't find a 3.3 (PSD ratio-) value in Figures 9 or 10 (in stage 3)

  Nahal Anim: There is a (PSD ratio-) value of approx. 1.3 for a water level between 0.4 and 0.5 m that you do not mention in the text.
- L408-409: Do high PSD ratios at low gyro velocities suggest that pebble vibrations contribute significantly to the generation of seismic energy, or are high PSD values (at low gyro velocities in Fig. 11) generally indicative of this?

**Specific comments on figures and tables**

**Figure 1:**

- What is the correct spelling of "Nahal Beer Sheva", with (as in the caption) or without a hyphen (as in the figure)? See also line 82.
- (a) In the inset, change spelling to "Mediterranean Sea"

**Figure 3:**

- Since abbreviations for events in the two rivers are introduced in lines 169–170, I strongly
  recommend that they be used consistently throughout the article. I would therefore change the
  caption as follows:
  - "(a-b) Nahal Anim, event A1; (c-d) Nahal Yatir, event Y1; (e-f) Nahal Yatir, event Y2; (g-h) Nahal Yatir, event Y3; (i-j) Nahal Yatir, event Y4."

**Figure 4:**

In the caption, change to
 "(a) Nahal Anim, event A1; (b-e) Nahal Yatir, events Y1-4."

**Figure 5:**

- Is there a reason, or is it explained anywhere, why Figure 5 only shows the events in the Nahal Yatir river and not the data for event A1 in the Nahal Anim river?
- In the caption (L222), change to "from Nahal Yatir (events Y1-4)."
- In the caption (L224), change to "the data is categorized"
- In the caption (L225-227): I wonder whether this explanation is sufficient for readers. This point is only mentioned in the captions for Figures 5, 8 and 9, but nowhere in the text. I leave that up to the authors, but I also wonder whether omitting the gyro data for the water level range from 0.08 to 0.25 m doesn't compromise the explanatory power of the graph.

**Figure 6:**

- I don't fully understand this figure. Shouldn't the percentages of all horizontal bars (for both states of motion) of the respective colour add up to 100%?
- In this context: What exactly is meant by "category"? The three stages (on the y-axis)? Or the six different fields when the stages are combined with the motion states?
- The caption for Figure 5 (and elsewhere in the article) uses the term "steady flow conditions" Here, however, the term "quasi-steady flow conditions" is used. Is there a valid reason for this? Otherwise, I suggest using consistent terminology.

**Figure 7:**

In the caption (L254), consider changing to "for the PSD-gyro velocity correlation"

**Figure 8:**

- In the caption (L265), change to "PSD against water depth during event A1, using the 35-60 Hz frequency band (a); and during events Y1-4, respectively using the 40–65, 60–85, 50–75, and 30–55 Hz frequency bands (b-e)."
- In the caption (L267), change to
   "PSD data for the water depth range of 0.08–0.25 m is not shown in (b-e), as..."

**Figure 9**

- I have difficulty seeing and understanding the distinction between Stage 2 and Stage 3 based on the data from Nahal Yatir in Figure 9b (and in the corresponding text in L275-277). The PSD ratios in these two (PSD based) stages seem quite similar.
- In the caption (L285-286), it would probably be good to mention that all events were used: "(a) data from Nahal Anim, event A1 and (b) from Nahal Yatir, events Y1-4."

**Figure 10**

- In the caption (L298), change to "PSD ratio against"
- In the caption (L298), change to "Nahal Anim event A1 (blue circles)"

**Figure 11**

• Caption (L310): I am quite confused by the introduction of this fourth (?) set of stages. Please also see my general comment on this matter. Doesn't the dashed line also represent a gyro velocity threshold? Here, it is set at 0.2 rad/s and is very close to the vibration-displacement threshold of 0.3 rad/s.

I am also not fully convinced by the argumentation for the definition of the dashed line in lines 305-306 of the text. I do not see the "distinct trends" described by the authors as being quite so clearly evident.

**Table 1:**

- Since abbreviations for events in the two rivers are introduced in lines 169–170, I strongly recommend that they be used consistently throughout the article. I would therefore change the caption as follows:
  - "across the five flow events in Nahal Anim (A1) and Nahal Yatir (Y1-Y4)."
- In addition to specifying the range (from to) of the rise in water level, it might be interesting to show the absolute increase in a column of Table 1.

**Technical corrections**

L168:

| L32:                   | Change to "glacial dam outbursts"                                                                                                                                                                                                                                                                                                                                                                                                                                                                                                                                                                                                                |
|------------------------|--------------------------------------------------------------------------------------------------------------------------------------------------------------------------------------------------------------------------------------------------------------------------------------------------------------------------------------------------------------------------------------------------------------------------------------------------------------------------------------------------------------------------------------------------------------------------------------------------------------------------------------------------|
| L32:                   | Change to "reservoir breaches"                                                                                                                                                                                                                                                                                                                                                                                                                                                                                                                                                                                                                   |
| L39:                   | Consider changing to: "Similarly, Meirovich et al. (1988) used measurements made in desert streams to demonstrate that"                                                                                                                                                                                                                                                                                                                                                                                                                                                                                                                          |
| L41:                   | Change to "can mobilize bedload even"                                                                                                                                                                                                                                                                                                                                                                                                                                                                                                                                                                                                            |
| L48-49:                | Consider changing to "These advances have the potential to greatly improve our"                                                                                                                                                                                                                                                                                                                                                                                                                                                                                                                                                                  |
| L63:                   | Please check the use of upper and lower case letters in the sentence "IMU are Micro-electro-mechanical Systems which"                                                                                                                                                                                                                                                                                                                                                                                                                                                                                                                            |
| L81:                   | Consider changing to "Both catchments are situated within the"                                                                                                                                                                                                                                                                                                                                                                                                                                                                                                                                                                                   |
| L83:                   | Change to "yet they have different grain sizes"                                                                                                                                                                                                                                                                                                                                                                                                                                                                                                                                                                                                  |
| L83-84:                | Please check this sentence. Do the authors mean: " bedload transport is influenced by morphological characteristics of the river bed under conditions of rapid stage increase."                                                                                                                                                                                                                                                                                                                                                                                                                                                                  |
|                        | conditions of rapid stage increase.                                                                                                                                                                                                                                                                                                                                                                                                                                                                                                                                                                                                              |
| L93:                   | Change to "was performed using"                                                                                                                                                                                                                                                                                                                                                                                                                                                                                                                                                                                                                  |
| L93:
L99:           |                                                                                                                                                                                                                                                                                                                                                                                                                                                                                                                                                                                                                                                  |
|                        | Change to "was performed using"                                                                                                                                                                                                                                                                                                                                                                                                                                                                                                                                                                                                                  |
| L99:                   | Change to "was performed using"  Change to "Additionally, we computed"  Consider changing to the following, or something similar, to avoid the double use of the word 'pebble': "An RFID tag was inserted into each pebble to help locate and identify                                                                                                                                                                                                                                                                                                                                                                                           |
| L99:
L110:          | Change to "was performed using"  Change to "Additionally, we computed"  Consider changing to the following, or something similar, to avoid the double use of the word 'pebble': "An RFID tag was inserted into each pebble to help locate and identify them."  A decision should be made as to whether the abbreviation for grain diameter ( D or d ) should be capitalised or lowercase throughout the text. Earlier in the text, on lines 86 and                                                                                                                                                                                 |
| L99:
L110:
L113: | Change to "was performed using"  Change to "Additionally, we computed"  Consider changing to the following, or something similar, to avoid the double use of the word 'pebble': "An RFID tag was inserted into each pebble to help locate and identify them."  A decision should be made as to whether the abbreviation for grain diameter ( $D$ or $d$ ) should be capitalised or lowercase throughout the text. Earlier in the text, on lines 86 and 88, " $D_{50}$ " is chosen.                                                                                                                                                               |
| L99:
L110:
L113: | Change to "was performed using"  Change to "Additionally, we computed"  Consider changing to the following, or something similar, to avoid the double use of the word 'pebble': "An RFID tag was inserted into each pebble to help locate and identify them."  A decision should be made as to whether the abbreviation for grain diameter ( D or d ) should be capitalised or lowercase throughout the text. Earlier in the text, on lines 86 and 88, " D 50 " is chosen.  Change to "measurements of Pretzlav et al. (2020), conducted with"  Choose between singular and plural here. My suggestion would be: |

Consider changing to "with a total of five events successfully recorded (Fig. 2),"

| L174:     | To avoid any mix-ups with the 'Max. rate of rise', consider changing to: "The highest rate of rise for an entire rapid increase" |
|-----------|----------------------------------------------------------------------------------------------------------------------------------|
| L203-204: | Change to "in events Y1-Y4" or "in events Y1-4"                                                                                  |
| L204:     | Should this refer to Figure 4 or Figure S4?                                                                                      |
| L230:     | Change to "the smartrock intermediate"                                                                                           |
| L240:     | Please check this subtitle. Do you mean "Characteristics of seismic energy"?                                                     |
| L247:     | Consider changing the subtitle to "PSD – smartrock gyro velocity correlation"                                                    |
| L251:     | Change to "for events Y1-4, with"                                                                                                |
| L259:     | Change to "ranging from -178"                                                                                                    |
| L281:     | Consider changing to "is characterized by PSD ratios exceeding unity,"                                                           |
| L329:     | Consider changing to "compared to steady flow conditions, extending to flow depths of $^{\sim}$ 0.10 m"                          |
| L345:     | In this subtitle, wouldn't "Transitional flow" be more accurate than "Transition"?                                               |
| L348:     | Change to "where the smartrock intermediate diameter approximates the"                                                           |
| L371:     | Change to "during steady flow,"                                                                                                  |
| L391-392: | I suggest splitting this sentence into two sentences and starting the second sentence with 'Thus, ratio values $\ldots$ '        |

**References**

L408:

The following articles referred to in the text are not included in the reference list and should be added there:

I would advise against referring to illustrations in the conclusions. This is rather unusual.

- Bilek (2019)
- Welch (1967)

The following articles are included in the list of references, but are not referenced in the text. They should be deleted from the list.

- Burtin et al. (2008)
- Burtin et al. (2014)